# A Contrastive Learning Pre-Training Method for Motif Occupancy Identification

**DOI:** 10.3390/ijms23094699

**Published:** 2022-04-24

**Authors:** Ken Lin, Xiongwen Quan, Wenya Yin, Han Zhang

**Affiliations:** College of Artificial Intelligence, Nankai University, Tianjin 300350, China; ken_lin@mail.nankai.edu.cn (K.L.); 2120210426@mail.nankai.edu.cn (W.Y.); zhanghan@nankai.edu.cn (H.Z.)

**Keywords:** contrastive learning, edit distance, data augmentation, sequence similarity, motif occupancy identification, pre-training

## Abstract

Motif occupancy identification is a binary classification task predicting the binding of DNA motif instances to transcription factors, for which several sequence-based methods have been proposed. However, through direct training, these end-to-end methods are lack of biological interpretability within their sequence representations. In this work, we propose a contrastive learning method to pre-train interpretable and robust DNA encoding for motif occupancy identification. We construct two alternative models to pre-train DNA sequential encoder, respectively: a self-supervised model and a supervised model. We augment the original sequences for contrastive learning with edit operations defined in edit distance. Specifically, we propose a sequence similarity criterion based on the Needleman–Wunsch algorithm to discriminate positive and negative sample pairs in self-supervised learning. Finally, a DNN classifier is fine-tuned along with the pre-trained encoder to predict the results of motif occupancy identification. Both proposed contrastive learning models outperform the baseline end-to-end CNN model and SimCLR method, reaching AUC of 0.811 and 0.823, respectively. Compared with the baseline method, our models show better robustness for small samples. Specifically, the self-supervised model is proved to be practicable in transfer learning.

## 1. Introduction

Transcription factor (TF) is a type of bio-functional proteins which can bind to a specialized DNA sequence. When binding to a DNA sequence, a TF controls the rate of transcription, where segments of the bound DNA containing genetic information is transcribed to messenger RNA [1,2]. The motif occupancy identification task is a binary classification task of predicting the binding of DNA motif instances to TF proteins [3]. Therefore, motif occupancy identification provides an important measure of binding affinity between DNA sequences and TFs.

In the past few years, several sequence-based methods for motif occupancy identification have been proposed. Ghandi et al. [4] proposed a gapped k-mer support-vector machine (gkm-SVM) based on SVM classifier and k-mer method, which combined conventional biology approach and machine learning for biology sequence analysis and function prediction. Zeng et al. [3] took advantage of the windowed kernel and deep learning characters of the convolutional neural network (CNN) to encode DNA sequences and predicted DNA-protein binding using a deep neural network (DNN). Based on the work of Zeng et al., Li et al. [5] introduced Bayesian method to the CNN model, preventing over-fitting problems of the original method. The methods above enhance the classification results benefiting from improved model structures. However, these methods are all direct training methods and their sequence encoders directly rely on the classification.

In this work, based on contrastive learning, we propose an interpretable and robust pre-training method for sequence encoders from the perspective of sequence representation and encoding. Our method aims at mining the implicit information of DNA sequences and acquiring helpful DNA representations for motif occupancy identification. Two main techniques applied in our work are contrastive learning and sequence edit similarity.

Contrastive learning is a branch of self-supervised learning methods and has been widely applied in computer vision field [6,7,8]. The pretext tasks developing from contrastive learning has also been used to self-supervised label generation and sequence encoding tasks in natural language processing, making it possible to process self-supervised data with supervised methods [9].

The most common type of contrastive learning is self-supervised contrastive learning, which defines similar pairs (positive samples) and dissimilar pairs (negative samples) from unlabeled dataset. Self-supervised learning methods try to get similar pairs closer and keep dissimilar pairs far away from each other in their representation space [10]. The representations are expected to be clustered in a self-supervised way. To discriminate similar samples from dissimilar samples, a variety of methods have been proposed, such as dictionary query [7] and contrast between augmented samples [8].

Moreover, the idea of contrastive learning has also been adopted on supervised data supporting classification, leading to supervised contrastive learning [11,12]. Compared to self-supervised contrastive learning, supervised contrastive learning introduces real label information about data and constructs correct positive and negative samples. Thus, supervised contrastive learning is able to discriminate misleading similar samples from different classes and vice versa.

Recently, contrastive learning has played a role in some biology sequence encoding tasks. Zou et al. [13] adopted two Hidden Markov Models to encode DNA sequences and applied contrastive analysis to DNA representations. Their results provide a powerful exploratory tool for biology. Wang et al. [14] combined contrastive learning with adversarial learning. They applied the auto-encoder to learn representations of single-cell RNA sequences and matched local distribution using the contrastive loss function, aiming at removing batch effects in single-cell RNA-seq data. Ciortan et al. [15] came up with a contrastive learning method for scRNA sequence encoding and clustering. They randomly masked each raw sequence to obtain two augmented sequences as a positive pair, and assigned sequences from different pairs as negative samples. Wan et al. [16] proposed a similar method which assigned the original sequence and the augmented one as a pair of positive samples. The works above show that contrastive learning is capable of encoding bio-macromolecular (especially genetic) sequences based on clustering and generating sequence representations for downstream classification.

In our method, to research on possible pre-training strategies of contrastive learning, two alternative models based, respectively, on self-supervised and supervised contrastive learning are constructed, which are different in contrastive strategies and possess respective advantages.

Another technique applied in our work is the sequence similarity based on edit distance. Raw DNA sequences are represented by character strings. We introduce a sequence similarity criterion for positive or negative sample assignment in contrastive learning. The similarity between character strings can be calculated based on edit distance. Given a source string $x_{i}$ and a target string $x_{j}$, edit distance quantifies the minimum number of edit operations to transform $x_{i}$ to $x_{j}$. The realization of edit distance is related to the definition and weight assignment of edit operations [17]. Some parameterized algorithms [18,19] and databases [20,21] for bio-sequence similarity based on edit distance have been applied to sequence alignment problems in bioinformatics.

In our method, we introduce a sequence similarity criterion for positive and negative DNA pair labeling in self-supervised contrastive learning based on the Needleman–Wunsch algorithm [18]. The algorithm is applicable to the computation of minimum edit distance as well as maximum similarity of two character strings such as a pair of DNA sequences.

## 2. Materials and Methods

### 2.1. Dataset and DNA Sequences

Our dataset is sampled from the datasets (accessed on 1 August 2020) constructed by Zeng et al. [3]. The full datasets are from Encyclopedia of DNA Elements (ENCODE) database [22] and consist of ChiP-seq experiments [23] on 422 TFs. We use the full datasets for some statistics about sequence similarity distribution searching for best threshold hyper-parameter setting. We sample 40 out of 422 datasets for model training and evaluation as contrastive learning requires much more time for training than end-to-end CNN model proposed by Zeng et al. [3].

All DNA sequences in the dataset are of same length, containing L=101 basic groups. The sequences are central sub-sequences of DNA motif instances, in which only positive samples overlap with ChIP-seq regions. In other words, the label of a DNA sequence is determined by whether it bind with any TF. In our work, the basic groups of DNA sequence are vectorized into 4-channel one-hot [24] encoding. Adenine (token *A*), thymine (token *T*), cytosine (token *C*), guanine (token *G*) are, respectively, represented by ${\left[1,0,0,0\right]}^{T}$, ${\left[0,1,0,0\right]}^{T}$, ${\left[0,0,1,0\right]}^{T}$ and ${\left[0,0,0,1\right]}^{T}$. The null token *N* (i.e., unknown basic group) is represented by ${\left[0,0,0,0\right]}^{T}$. The input DNA sequences are encoded as L×4 tensors.

### 2.2. Model Overview

To explore possible pre-training strategies with our method, we design two alternative contrastive learning for representation (CLR) models: self-supervised editCLR and supervised supCLR, both with the same structure of DNA sequence encoder and feature classification shown in Figure 1. The differences between the two models lie in their strategy of contrastive sample labeling and loss function definition, and their unique characteristics are, respectively, elaborated in Section 2.5 and Section 2.6. The structures of the CNN encoder and DNN classifier refer to the DeepSEA [25] and DeepBind [26] architecture.

The encoder is composed of a convolutional layer and a global max pooling layer. The convolutional layer extracts local sub-sequential information of input sequence *x* with fixed-sized kernels. The stride of convolution is set to 1 to traverse the whole sub-sequential space. Multiple convolutional kernels with different weights extract windowed features from a variety of perspectives. Each channel of feature vector will be filtered into one most important numerical value by global max pooling and concatenated with other channel digits into an encoded feature *z*. The DNN classifier consists of two full connection layers and one Dropout layer between them [27].

For a single DNA sequence, our referred end-to-end methods encode it with a CNN encoder and directly classify it by the encoded features. Although these methods are straightforward, they are lack of biological interpretability. In this work, we apply the thinking of contrastive learning and design a strategy for encoder pre-training with a contrastive loss function. Pre-trained by our contrastive learning method, the CNN encoder learns inherent and prior rules about motif occupancy identification and helps the classifier with final inference.

We introduce our detailed methods in the following sections. Section 2.3 and Section 2.4 are about sequence similarity computation and data pre-processing. Section 2.5 illustrates our self-supervised model editCLR. Section 2.6 illustrates our supervised model supCLR. Finally, Section 2.7 discusses about our hyper-parameter settings. For better readability, we list the meanings of important symbols in Table 1.

### 2.3. The Needleman–Wunsch Algorithm and Sequence Similarity

Inspired by the common biological (especially genetic) sequence alignment methods [18,19] and databases [20,21], we introduce the biological interpretability of sequence similarity based on edit distance and the Needleman–Wunsch algorithm.

The Needleman–Wunsch algorithm is a dynamic programming algorithm based on edit distance for sequence alignment [18]. The three key parameters of the algorithm are Match, Mismatch and Indel score which determine the actual meaning of the results. In our setting, the result about similarity scores 1 per Match and 0 per Mismatch and Indel.

The lengths of DNA sequences in our datasets are all same. Given two *L*-length DNA sequences $x_{i}$ and $x_{j}$, their Needleman–Wunsch score $F_{LL}\left(x_{i},x_{j}\right)$ (abbreviated as $F_{LL}$ function) is obtained by the iterative Equation (Equation 1) according to our parameter settings:(1)$$F_{pq}=\max\left\{F_{p-1,q-1}+chsim\left(x_{i}^{p},x_{j}^{q}\right),F_{p-1,q},F_{p,q-1}\right\},$$
where *p* denotes the index of $x_{i}$ and $x_{i}^{p}$ denotes the *p*-th character of $x_{i}$. Similarly, *q* is the index of $x_{j}$ and $x_{j}^{q}$ is the *q*-th character of $x_{j}$. The character matching score $chsim\left(x_{i}^{p},x_{j}^{q}\right)$ is defined by Equation (Equation 2).
(2)$$chsim\left(x_{i}^{p},x_{j}^{q}\right)=\left\{\begin{array}{cc} 1, & x_{i}^{p}=x_{j}^{q}\\ 0, & x_{i}^{p}\neq x_{j}^{q} \end{array}\right..$$

Equation (Equation 1) indicates that the Needleman–Wunsch score between $x_{i}$ and $x_{j}$ reaches the maximum *L* only if they are identical. In addition, the minimum score is 0 only when the deoxyribonucleotide types in two sequences are entirely different. Therefore we define the similarity between two DNA strings $xsim\left(x_{i},x_{j}\right)$ be their Needleman–Wunsch score divided by their common length *L*, that is Equation (Equation 3).
(3)xsimxi,xj=FLLxi,xjFLLxi,xjLL.

### 2.4. Sequence Edit Augmentation

Data augmentation [28,29] is a mainly used method for positive and negative sample generation in self-supervised contrastive learning. The famous SimCLR model [8] augmented the original data in two ways and obtained a pair of samples. The samples in the same pair constitute a positive sample and those in different pairs are negative samples.

In our method, we augment our original DNA sequences by insertion, deletion and substitution, which are edit operations defined by the Needleman–Wunsch algorithm. To keep the parametric computing stable, we ensure the augmented sequences are in the same length. The similarity (calculated by Equation (Equation 3)) between two augmented samples of one original sequence is guaranteed to be larger than 90%.

As illustrated in Figure 2, for insertion operations, we insert random tokens at random positions. Each insertion corresponds to a deletion operation to maintain the sequence length. A substitution randomly replaces a token with another one and an individual deletion is to replace a token with null token *N*. For a mini-batch data of size *n*, we acquire 2n augmented sequences.

### 2.5. Self-Supervised Contrastive Learning Based on a Sequence Similarity Criterion

We first construct a self-supervised contrastive learning model, named editCLR. In editCLR, our criterion of sample pair labeling is not simply based on the original sequences of augmented samples, but judging from their similarities.

A sequence similarity criterion with two thresholds is proposed to determine the attributes of sample pairs. Given a pair of samples, if their similarity is greater than the positive threshold, they are treated as a positive pair. Or if the similarity is less than the negative threshold, they are judged as a negative pair. Specifically, the quantity demand of negative pairs is much higher than that of positive pairs in contrastive learning; thus, we avoid the circumstance that a pair with low similarity becomes a positive one. We assign the thresholds with the following standard: if a similarity lies in the interval where the ratio of positive pairs is not significant and the variance of distribution is high, it must be between the positive and negative thresholds, where the pair is regarded as a neutral one. The neutral pairs will not participant in the loss calculation in self-supervised contrastive learning.

To explore the relationship between sequence similarities and classification results, we randomly sample 0.01% of possible sequence pairs (about $1.83\times 10^{8}$ unique combinations without order) in every dataset. According to the statistical results, the proportion of positive to negative pairs is almost 1:1. For each dataset, we count the minimum and maximum similarities of all sampled pairs and average similarities of positive pairs (i.e., with a same supervised label) and negative pairs (i.e., with different labels). As shown in Figure 3, the similarities of sampled pairs distribute between 0.3 to 1.0. Both average similarities of positive pairs and negative pairs are about 0.6 and the results of all datasets are nearly consistent.

To obtain the accurate thresholds, we count the sampled pairs in each similarity intervals (the width of an interval is 0.1). Figure 4 shows that the similarities of most pairs (about 99.3%) are concentrated in intervals between 0.5 and 0.7. We have discussed that positive pairs make up about 50% of pairs in these intervals. However, the variance of pair counts is not high due to the concentration. Therefore, these pairs are treated as negative pairs in self-supervised learning, offering abundant negative samples for model training [8].

We furthermore count the amount of positive and negative pairs by similarity intervals and calculate the proportion of positive pairs, yielding Figure 5. In the intervals with similarities less than 0.4, the median of positive pair proportion is smaller than 50%. For pairs in similarity over 0.7, the median of positive pair is larger than 50% and grows with similarity. Specifically, in similarity interval 0.7 to 0.8, the distribution of positive pairs is scattered, according with our threshold standard of neutral samples. Finally, the negative threshold is set as 0.7 and positive threshold is 0.8.

Figure 5 also uncovers the evidence that motif occupancy results of a pair of DNA sequences are related with their similarities. This means the sequence similarity can be a prior guidance on the motif occupancy identification.

As a result, we specify that, given a pair of augmented DNA sequences $x_{i1}$ and $x_{j2}$, the self-supervised contrastive learning label is assigned by Equation (Equation 4).
(4)$$label^{self}\left(x_{i1},x_{j2}\right)=\left\{\begin{array}{cc} 1, & xsim\left(x_{i1},x_{j2}\right)>0.7\\ 0, & xsim\left(x_{i1},x_{j2}\right)<0.4\\ -, & others \end{array}\right.,$$
where − represents that no label is assigned to neutral pairs.

The participation of sequence similarity criterion offers a stricter constraint on the assignment of negative pairs in contrastive learning and properly adjusts the number of positive pairs, making it easier for model to encode sequence helpful for downstream classification.

Our self-supervised contrastive loss function is in the same form with that of SimCLR [8], but we need to adjust the function due to absent neutral pairs. The definition of our self-supervised contrastive loss function is as follows:

Given an *n*-sized mini batch data $\left\{x_{1},x_{2},\cdots ,x_{n}\right\}$, 2n augmented samples are generated by edit operations. The *i*-th original sequence $x_{i}$ produces $x_{i1}$ and $x_{i2}$, which are encoded into $z_{i1}$ and $z_{i2}$, respectively, through the shared CNN encoder. As defined in Equation (Equation 5), the total self-supervised contrastive loss $L^{self}$ is the summation of each augmented sample loss.
(5)$$L^{self}=\sum_{i=1}^{n}\left(L_{i1}^{self}+L_{i2}^{self}\right).$$

Let $I\equiv \left\{x_{11},x_{12},x_{21},x_{22},\ldots ,x_{n1},x_{n2}\right\}$ be the set of all augmented samples. For an augmented sample $x_{i1}$, let $A\left(i1\right)\equiv I\backslash \left\{x_{i1}\right\}$ be other samples in the full set. Let $P\left(i1\right)\equiv \left\{x_{p}\in A\left(i1\right),label^{self}\left(x_{i1},x_{p}\right)=1\right\}$ be the positive set where samples can be judged as positive pairs with $x_{i1}$, $N\left(i1\right)\equiv \left\{x_{p}\in A\left(i1\right),label^{self}\left(x_{i1},x_{p}\right)=0\right\}$ be the negative set with $x_{i1}$. Then the sample loss of $x_{i1}$, named $L_{i1}^{self}$, is calculated by Equation (Equation 6).
(6)Li1self=−Pi1+Ni1Ai1Pi1∑xp∈Pi1logexpzsimzi1,zpzsimzi1,zpττ∑xq∈Pi1∪Ni1expzsimzi1,zqzsimzi1,zqττ,
where τ is a constant temperature hyper-parameter and zsim denotes the cosine similarity as Equation (Equation 7).
(7)$$zsim\left(z_{i1},z_{p}\right)=\frac{z_{i1}\cdot z_{p}}{∥z_{i1}∥∥z_{p}∥}.$$

Compared with the loss function of SimCLR, the main modification of Equation (Equation 6) is the scaling parameter $\frac{\left|P\left(i1\right)\right|+\left|N\left(i1\right)\right|}{\left|A\left(i1\right)\right|\left|P\left(i1\right)\right|}$. We exclude neutral pairs of samples (i.e., the samples in set $A\backslash \left(P\cup N\right)$) in our self-supervised contrastive learning process. The sizes of positive and negative sets are variable for every augmented sample and might lead to the instability of loss values and the oscillation of gradients in the neural network. We add a scaling parameter to adjust the loss values within a fixed scale of single positive pair (and $\left|A\right|-1=2n-2$ negative pairs) loss.

### 2.6. Supervised Contrastive Learning

We construct another supervised contrastive learning model, called supCLR, as an alternative to our editCLR pre-training model. The idea of supCLR is more transductive than that of editCLR, that is, directly employing the information of supervised labels. As mentioned in the research on supervised learning of Khosla et al. [11], supervised contrastive learning extends the assignment of positive pairs from a same original data to a same class. In this circumstance, supervised information becomes strong constraints on contrastive learning. From the view of clustering, supervised contrastive learning offers some interpretability compared with direct classification.

In our supCLR model, augmented samples are also transformed by edit operations, in which only positive pairs are from original sequences with a same supervised label. Neutral pairs no longer exist in supervised contrastive learning.

Figure 5 shows that sampled pairs with high similarities (e.g., over 0.9) are most in the same class globally, but there are still a few but not very few exceptions. As described in Section 2.2, the fixed window property of convolution trends to encode highly similar samples to be very close in the representation space, which may let the confusing exceptions mislead the model. We want to explore the effect of supervised information on these exceptions and experiment on whether supervised contrastive learning could improve sequence representations.

We redefine the contrastive label of sample pairs. Given a pair of augmented DNA sequences $x_{i1}$ and $x_{j2}$ with original supervised labels $y_{i}$ and $y_{j}$, respectively, their label for supervised contrastive is obtained by Equation (Equation 8).
(8)$$label^{sup}\left(x_{i1},x_{j2}\right)=\left\{\begin{array}{cc} 1, & y_{i}=y_{j}\\ 0, & y_{i}\neq y_{j} \end{array}\right..$$

Same with Equation (Equation 5), the total supervised contrastive loss $L^{sup}$ defined in Equation (Equation 9) sums up each sample loss:(9)$$L^{sup}=\sum_{i=1}^{n}\left(L_{i1}^{sup}+L_{i2}^{sup}\right).$$

For an augmented sample $x_{i1}$, we redefine its positive set for our supCLR model by $P\left(i1\right)\equiv \left\{x_{p}\in A\left(i1\right),label^{sup}\left(x_{i1},x_{p}\right)=1\right\}$. In supervised contrastive learning, we do not discard any pair; therefore, the negative set is just $N\left(i1\right)\equiv A\left(i1\right)\backslash P\left(i1\right)$. Although the number of samples is settled, the size of positive set is still a variable in connection with $x_{i1}$. Thus the scaling parameter is a function of positive set size and $x_{i1}$’s supervised contrastive sample loss $L_{i1}^{sup}$ is defined by Equation (Equation 10).
(10)Li1sup=−1Pi1∑xp∈Pi1logexpzsimzi1,zpzsimzi1,zpττ∑xq∈Ai1expzsimzi1,zqzsimzi1,zqττ.

### 2.7. Model Training and Hyper-Parameter Settings

The datasets introduced in Section 2.1 contain training sets and test sets. Referring to the work of Zeng et al. [3], we randomly sample 1/8 from the training sets as validation sets for selection of some hyper-parameters and the final models. Our training procedure can be divided into two steps. In the first step, we pre-train the CNN encoder by minimizing the contrastive loss function. In the second step, we fine-tune both the CNN encoder and the DNN classifier for downstream prediction. The self-supervised editCLR and supervised supCLR are trained individually.

As a reference of the DeepSEA and DeepBind, we set the kernel size of convolution as 24, the number of kernels as 64, and the number of hidden units in DNN as 32. We apply some automatic hyper-parameter grid search methods to find the best Dropout rate (searching within 0.1, 0.25, 0.5), temperature parameter τ (searching within 1.0, 0.1, 0.01) and some other parameters mentioned below.

The structural hyper-parameters in editCLR and supCLR are the same. However, the proportion of positive pairs in supCLR is much larger than that in editCLR, making differences of some hyper-parameters between the two models.

In pre-training, we apply mini-batch training with batch size n=1024 as a trade-off between the proportion of negative pairs in contrastive learning and the $O\left(n^{2}\right)$ time complexity for training. We adopt the Adam optimizer [30] to minimize the contrastive loss. The recommended learning rate of SimCLR [8] is $0.075\sqrt{n}$, we set our learning rate with a variable as $\alpha \sqrt{n}$, where α is searched logarithmically from 0.001 to 0.5 and α of editCLR might be smaller. The weight decay of the optimizer is searched logarithmically from 0.00001 to 0.01 and might be larger in editCLR. We set the maximum training epoch as 5.

In fine-tuning, the batch size is 128 for training. We adopt another Adam optimizer with both learning rate and weight decay as 0.001 to minimize a binary cross-entropy loss function [31]. The maximum epoch in fine-tuning is 20. Finally, we select the model of an epoch with the highest accuracy.

Our models are pre-trained and fine-tuned on an NVIDIA GeForece RTX 2070 GPU. For each iteration of a batch in pre-training, the computation of sequence similarities cost about 4.1 s. Provided better computation abilities, larger pre-training batch size might enhance the performance of the encoder.

## 3. Results

### 3.1. Comparison Experiments and Performance Measure

We construct an end-to-end CNN model without using contrastive learning as the baseline model. The encoder and the classifier of this baseline model are totally same with our editCLR and supCLR models, as well as their common hyper-parameters.

We measure the performance of all methods by accuracy, precision, recall, $F_{1}$ score and AUC. Through Softmax transformation, the 2-dimensional output of the classifier transforms into two probability values. We use $y_{pred}$ to denote the second digit, that is the predicted TF binding probability of the input DNA sequence (or the motif instance it belongs to). The model predicts a sequence to be a positive sample if $y_{pred}>0.5$. Besides the straightforward accuracy, we calculate the precision *P*, recall *R* and $F_{1}$ score [32] of the binary classification results by Equations (Equation 11)–(Equation 13), respectively.
(11)$$P=\frac{TP}{TP+FP}$$
(12)$$R=\frac{TP}{TP+FN}$$
(13)$$F_{1}=\frac{TP}{TP+\frac{1}{2}\left(FP+FN\right)},$$
where TP, FP, FN denote True Positive (both label and prediction are positive), False Positive (positive prediction for negative label) and False Negative (negative prediction for positive label), respectively.

Additionally, by changing the prediction threshold we calculate area (area under curve, AUC) under the receiver operating characteristic (ROC) curve [33] between False Positive Rate FPR (the proportion of False Positive to all negative samples) and True Positive Rate TPR (the proportion of True Positive to all positive samples).

### 3.2. Comparison Results

Firstly, we compare our editCLR and supCLR methods with the baseline CNN model deployed as Section 2.2 and the SimCLR model proposed by Chen et al. [8]. Specifically, we compare with two extra editCLR models without assigning neutral pairs (i.e., positive threshold equals to negative threshold). We name editCLR model with 0.7 thresholds as editCLR-0.7 and another with 0.8 thresholds as editCLR-0.8. The average results of accuracy, precision, recall, $F_{1}$ score and AUC of 40 datasets are listed in Table 2.

As shown in Table 2, both editCLR and supCLR methods outperform the baseline model in all measures and show general better performance than the SimCLR method, which means that our contrastive learning method is qualified to improve the prediction results of motif occupancy identification. Our editCLR models without neutral pairs perform close to SimCLR but obtain inferior results to standard editCLR model, which proves that the neutral pairs improve the effect of our self-supervised contrastive learning model. And supCLR is superior to all other compared methods on four measures, indicating the outstanding performance of supervised contrastive learning.

In detail, we illustrate the AUC distribution by dataset of all compared methods in Figure 6. The figure shows that the supervised supCLR preforms generally better than other methods and makes more stable and precise predictions. All contrastive learning based models including SimCLR gain relative better results than CNN baseline model, proving the advantages of contrastive learning over direct learning for motif occupancy identification. Our self-supervised editCLR model is similar with the SimCLR, but the introduction of the edit similarity criterion becomes one of editCLR’s advantages over SimCLR.

### 3.3. Analysis of Small Sample Learning

To study the robustness of our models, we furthermore experiment on the baseline model, SimCLR, editCLR and supCLR with smaller training ratio. We, respectively, sample 50%, 20% and 10% of the training sets to train the three models and evaluate them on the original test sets. The results are listed in Table 3 and the average results of AUC with different training ratios are illustrated in Figure 7.

The results in Table 3 indicate that even with smaller training size, our contrastive learning method is robust for motif occupancy identification. In addition, all the pre-training models are less sensitive to the training size than the baseline direct learning model. This is probably due to the difference in quantity of information obtained by the encoders, where pre-training task offers adequate sample pairs but direct learning does not. In addition, further benefiting from the nature of sequence similarity, self-supervised editCLR becomes the most adaptive model for small sample learning.

As shown in Figure 7 and the results, the baseline model suffers from over-fitting when the training set size is large. However, with a small training sample, the baseline model is not robust enough to overcome the under-fitting problem. Our self-supervised method overcomes over-fitting problem by introducing the general relationships between sequence similarity and classification results, which are statistically certain according to our statistics in Section 2.5. In addition, our supervised method makes sure that the encoder directly and adequately learns to pre-processes the features with supervised information. This helps the model to learn better representations and in some ways alleviates over-fitting. Therefore, our method possesses the advantage of robustness over different training sizes.

As mentioned in Section 3.2, our editCLR is similar with SimCLR but performs slightly better than it. Figure 7 also indicates the robustness of SimCLR on small sample training and the performance gap between SimCLR and our method.

### 3.4. Self-Supervised Model for Transfer Learning

We separate the model training procedure into pre-training and fine-tuning. In the self-supervised editCLR pre-training model, CNN encoder learns the representations of DNA sequences without supervised information. This means that editCLR encodes DNA sequences with their implicit and inherent information, and might be applied to transfer learning.

Our baseline model is one of the researched architectures in the work of Zeng et al. [3], named 1layer_64motif for 1 layer and 64 hidden-sized CNN encoder. The best architectures for motif occupancy identification in the above work are 1layer_64motif and 1layer_128motif (with 128 hidden units in CNN encoder), so we select a few pre-trained editCLR models on both 1layer_64motif and 1layer_128motif architectures from datasets with no less than 100,000 samples in training set and fine-tune them on other 20 datasets. As shown in Table 4, model examples of both two architectures perform comparable with the CNN baseline model even through they are transferred to other datasets, showing the robustness of our editCLR method in transfer learning.

## 4. Discussion

Motif occupancy identification is a binary classification task of predicting the binding of DNA motif instances to TF proteins, which is important for research on genetics and bioinformatics. Several sequence-based methods with computational machine learning or deep learning models for motif occupancy identification have been proposed. However, these methods are all end-to-end designs and make prediction for motif occupancy identification directly, which are straightforward but lack of biological interpretability. Moreover, these methods are not adaptive or robust enough for transfer learning and small sample learning.

In this work, we propose a contrastive learning method for DNA encoding and motif occupancy identification, along with our design of sequence similarity criterion and modified contrastive loss function. We construct two models to pre-train DNA sequential encoder: a self-supervised contrastive learning model editCLR based on sequence edit similarity criterion and a supervised contrastive learning model supCLR based on sequence supervised labels. We augment the original sequences for contrastive learning with edit operations defined in edit distance. Specifically, we propose a sequence similarity criterion based on the Needleman–Wunsch algorithm to discriminate positive and negative sample pairs and weed out controversial sample pairs in self-supervised contrastive learning. Finally, a simple DNN classifier is fine-tuned along with the pre-trained encoder to predict the results of motif occupancy identification.

Compared with the end-to-end model with a same structure, our self-supervised editCLR and supervised supCLR show better performance in motif occupancy identification. Both model are robust with small samples and the self-supervised editCLR can also be applied in transfer learning.

There are still some limitations of our method. As shown in Figure 6, a few outlier points appear in the results of editCLR and supCLR models. These points might result from the lack of pre-training epochs or bias in thresholds for a few datasets. This exposes a problem about hyper-parameter settings, that is, for a single dataset, the best hyper-parameters are hard to determine. This problem might be solved by introducing soft-value labels [34,35] to the loss function.

Another limitation is the cost of the similarity computing in self-supervised contrastive learning. In our pre-training model, the Needleman–Wunsch algorithm is implemented on a batch with the computational complexity of $O\left(n^{2}L^{2}\right)$. Some faster similarity estimation methods and parallel computing techniques [36,37,38] might be helpful for this problem. For further works, we will consider overcoming the limitations above.

## 5. Conclusions

Motif occupancy identification task predicts whether DNA motif instances bind to any transcription factor proteins. In this work, we propose a powerful and robust contrastive learning method and two types of alternative models for sequence encoding and motif occupancy identification. Experiments and multiple measures show that both our contrastive models outperform the end-to-end CNN model using direct learning, and make better prediction than SimCLR contrastive method. We also compare all methods on small training samples and find better robustness of our proposed method. Specifically, the supervised model provides significance enhancement of the results on almost all measures and the self-supervised model shows comparable outcomes for transfer learning by catching inherit relations between sequence similarities and classification label types.

## Figures and Tables

**Figure 1 ijms-23-04699-f001:**
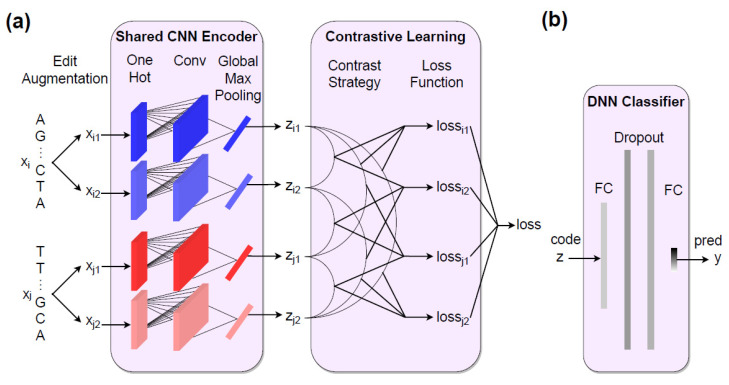
An overview of our (**a**) CNN encoder and contrastive learning procedure and (**b**) DNN classifier. Sequences ’AG...CTA’ and ’TT...GCA’ are examples of a mini-batch input raw DNA sequences. Conv denotes a convolution layer. FC denotes a full connection layer. The mark ’code’ represents encoded features and ’pred’ represents output prediction scores.

**Figure 2 ijms-23-04699-f002:**
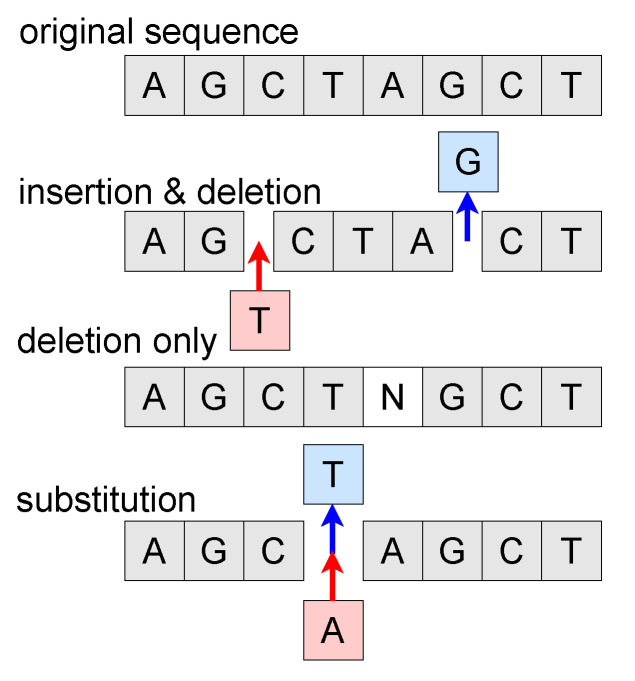
An illustration of three DNA sequence edit augmentation ways. Tokens *A*, *G*, *C*, *T* denote four types of deoxyribonucleotides and token *N* denotes unknown basic group or null token. Red arrows denote inserting and blue ones denote removing.

**Figure 3 ijms-23-04699-f003:**
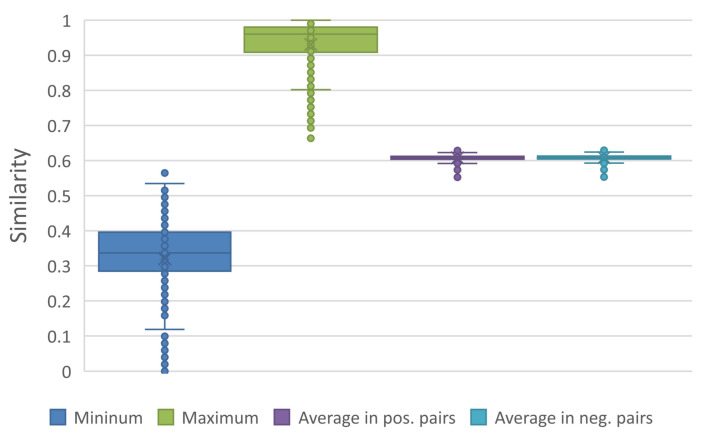
Maximum, minimum and average similarities in all datasets. Abbreviation pos. (positive) means the pairs with a same supervised label, and neg. (negative) means those with different label (similarly hereinafter).

**Figure 4 ijms-23-04699-f004:**
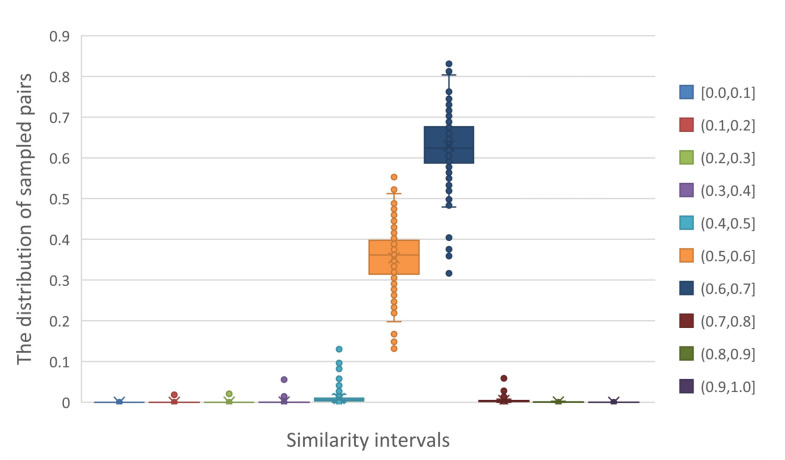
The distribution of sampled pairs on each similarity interval in all datasets.

**Figure 5 ijms-23-04699-f005:**
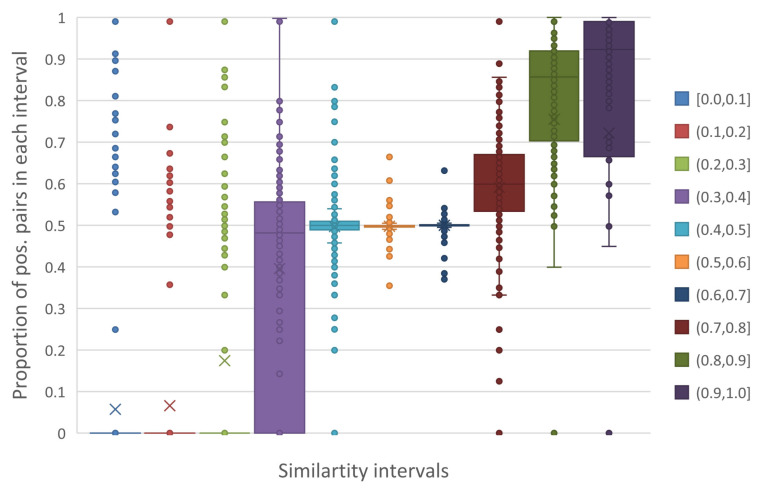
The proportion of positive pairs in each similarity interval in all datasets.

**Figure 6 ijms-23-04699-f006:**
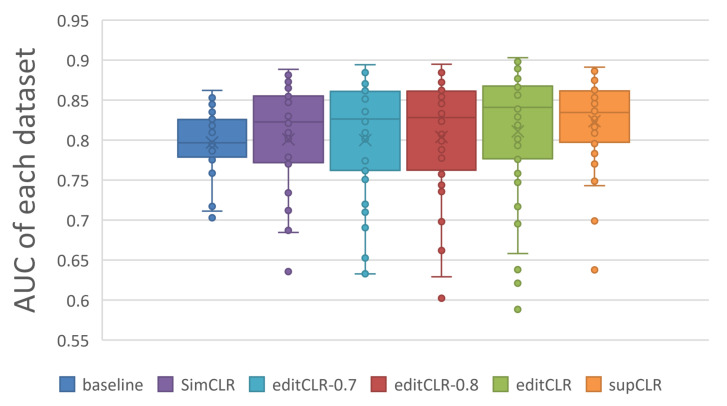
The distribution of AUC results on all datasets.

**Figure 7 ijms-23-04699-f007:**
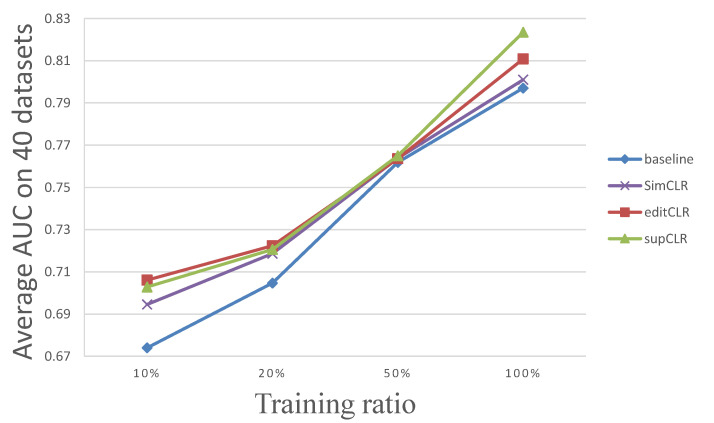
The distribution of AUC results on all datasets with different training ratios.

**Table 1 ijms-23-04699-t001:** A glossary of important symbols in this section.

Type	Symbol	Meaning
Variables	$x,x_{i},x_{j}$	input sequences
	$y_{i},y_{j}$	classification labels of input sequences
	$x_{i}^{p}$	*p*-th character of $x_{i}$
	$x_{i1},x_{i2}$	augmented sequences of $x_{i}$
	$z,z_{i1},z_{i2}$	encodes features of augmented sequences
	τ	constant temperature hyper-parameter
	α	zoom ratio hyper-parameter of learning rate
	$L^{self}$	total self-supervised contrastive loss
	$L^{sup}$	total supervised contrastive loss
	$L_{i1}^{self}$	sample self-supervised loss of $x_{i1}$
	$L_{i1}^{sup}$	sample supervised loss of $x_{i1}$
Sets	*I*	full augmented sample set
	$A\left(i1\right),A$	augmented sample set excluding $x_{i1}$
	$P\left(i1\right),P$	positive sample set of $x_{i1}$
	$N\left(i1\right),N$	negative sample set of $x_{i1}$
Functions	$F_{pq}\left(x_{i},x_{j}\right)$	Needleman–Wunsch score between first *p* characters of $x_{i}$
		and first *q* characters of $x_{j}$
	$chsim\left(x_{i}^{p},x_{j}^{q}\right)$	character matching score between characters $x_{i}^{p}$ and $x_{j}^{q}$
	$xsim\left(x_{i},x_{j}\right)$	similarity score between sequences $x_{i}$ and $x_{j}$
	$zsim\left(z_{i1},z_{p}\right)$	cosine similarity score between features $z_{i1}$ and $z_{p}$
	$label^{self}\left(x_{i1},x_{j2}\right)$	self-supervised similarity label between sequences $x_{i1}$ and $x_{j2}$
	$label^{sup}\left(x_{i1},x_{j2}\right)$	supervised similarity label between sequences $x_{i1}$ and $x_{j2}$

**Table 2 ijms-23-04699-t002:** Performance results on DEEPre dataset.

Method	Accuracy	$F_{1}$ Score	Precision	Recall	AUC
baseline	0.716	0.702	0.738	0.675	0.797
simCLR	0.719	0.721	0.761	0.689	0.801
editCLR-0.7	0.720	0.721	0.760	0.692	0.800
editCLR-0.8	0.720	0.722	0.760	0.690	0.804
editCLR	0.723	0.725	0.762	**0.694**	0.811
supCLR	**0.734**	**0.734**	**0.784**	0.691	**0.823**

All results are average values of 40 datasets. Bold values represent the best results for corresponding measures (similarly hereinafter).

**Table 3 ijms-23-04699-t003:** Performance results on DEEPre dataset.

Method	Training Ratio	Accuracy	$F_{1}$ Score	AUC
baseline	50%	0.688	0.683	0.762
SimCLR		0.690	0.697	0.764
editCLR		0.693	**0.700**	0.764
supCLR		**0.700**	0.698	**0.765**
baseline	20%	0.636	0.632	0.705
SimCLR		0.653	0.659	0.719
editCLR		0.657	**0.664**	**0.722**
supCLR		**0.658**	0.656	0.721
baseline	10%	0.629	0.607	0.674
SimCLR		0.636	0.646	0.695
editCLR		**0.640**	**0.650**	**0.706**
supCLR		0.639	0.645	0.703

All results are average values of 40 datasets.

**Table 4 ijms-23-04699-t004:** The average results of 8 sampled pre-trained editCLR models transferred to 20 datasets.

Architecture	Model	Accuracy	$F_{1}$ Score	AUC
1layer64motif	baseline	0.717	0.707	0.794
	editCLR#1	0.693	0.718	0.768
	editCLR#2	**0.720**	0.715	**0.807**
	editCLR#3	0.715	**0.729**	0.792
	editCLR#4	0.715	0.711	0.806
1layer128motif	baseline	0.723	0.712	0.805
	editCLR#5	0.725	**0.727**	**0.811**
	editCLR#6	**0.726**	0.715	0.804
	editCLR#7	0.718	0.709	0.800
	editCLR#8	0.721	0.716	0.808

All results are average values of 20 datasets which are not the original dataset where editCLR models are pre-trained. The baseline model is directly trained on the 20 datasets without transfer learning.

## Data Availability

All datasets presented in this study (see Section 2.1) are openly available at http://cnn.csail.mit.edu/deepbind_pred/ and accessed by this work on 1 August 2020.

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
