# Peer review of "A Contrastive Learning Pre-Training Method for Motif Occupancy Identification"

_ijms, 2022, doi:10.3390/ijms23094699_

Round 1

Reviewer 1 Report

1.There is no discussion on the cost effectiveness of the proposed method. What is the computational complexity? What is the runtime? Please include such discussions.

2.Please state how to fix overfitting in your CNN model?

3.There is no discussion on the confusion matrix. Please state confusion matrix.

4.Please well define mathematical symbols and abbreviations in this manuscript.

5.Please provide more explanations of the conclusions from corresponding studies.

6.Please compare with more methods to clarify the powerful and robust of the approach.

7.Please try to refer the latest references.

Reviewer 2 Report

This paper by Lin et al. reported a contrastive learning method to pre-train interpretable and robust DNA encoding for motif occupancy identification. The DNA sequential encoder was pre-trained with two alternative models - a self-supervised model and a supervised model. In their self-supervised learning, the authors proposed a sequence similarity criterion base on the Needleman-Wunsch algorithm to enable positive/negative sample pairs discrimination, and DNN model architecture was introduced as well. The authors’ conclusions show that their model has better performance metrics compared to the existing end-to-end CNN model. I think the topic of this work is of good importance and interest, and I only have a few comments listed below that I think needed to be further addressed before I can recommend its publication to IJMS.

  1. How the hyperparameters were tuned? Has the technics like grid search been applied? How was the prevention of over-fitting been realized? Are there any regularization terms other than the dropout technics?
  2. What is the ratio of the neutral pairs after this definition? It seems that the choice of picking threshold is very vital, could the authors validate their way of threshold assignment and show the robustness of the conclusions? For example, what if the threshold was set to different values? What would happen? What if the neutral pairs are not allowed?
  3. Though indeed improved, the improvement brought by the model does not seem to be very significant. If I understand correctly, a lot of model architectures are directly inherited from the existing model that the authors’ model was built upon. I wonder if the authors could explore the opportunity of any further enhancement of the model performance by searching over the architectural alternations (from the simples hyperparameters like the kernel size and the number of hidden units). The current results serve as a good benchmark but it would be interesting to do the model optimization in that sense.
  4. Can the authors provide more in-depth discussions on the possible reasons that their model is outperforming other existing models? For example, is it mainly because of the novel introduction of the sequence similarity criterion based on the Needleman-Wunsch algorithm with augmented sequences? And can the authors provide more mechanistic interpretations on why the model is more robust on small samples compared to the end-to-end model with the same structures?
  5. Also, the authors claim that their model is useful in the application of transfer learning. It would be useful to show at least one or two quick examples to support their claim.

Round 2

Reviewer 1 Report

Thanks for your response.

Reviewer 2 Report

The authors have addressed my concerns and the manuscript has been largely improved. I can recommend its publication to IJMS.